# Biomolecular interactions modulate macromolecular structure and dynamics in atomistic model of a bacterial cytoplasm

Isseki Yu[1,2], Takaharu Mori[1,2], Tadashi Ando[3], Ryuhei Harada[4], Jaewoon Jung[4], Yuji Sugita[1,2,3,4]*, Michael Feig[3,5]*

[1]iTHES Research Group, RIKEN, Saitama, Japan; [2]Theoretical Molecular Science Laboratory, RIKEN, Saitama, Japan; [3]Laboratory for Biomolecular Function Simulation, RIKEN Quantitative Biology Center, Kobe, Japan; [4]Computational Biophysics Research Team, RIKEN Advanced Institute for Computational Science, Kobe, Japan; [5]Department of Biochemistry and Molecular Biology, Michigan State University, East Lansing, United States

**Abstract** Biological macromolecules function in highly crowded cellular environments. The structure and dynamics of proteins and nucleic acids are well characterized in vitro, but in vivo crowding effects remain unclear. Using molecular dynamics simulations of a comprehensive atomistic model cytoplasm we found that protein-protein interactions may destabilize native protein structures, whereas metabolite interactions may induce more compact states due to electrostatic screening. Protein-protein interactions also resulted in significant variations in reduced macromolecular diffusion under crowded conditions, while metabolites exhibited significant two-dimensional surface diffusion and altered protein-ligand binding that may reduce the effective concentration of metabolites and ligands in vivo. Metabolic enzymes showed weak non-specific association in cellular environments attributed to solvation and entropic effects. These effects are expected to have broad implications for the in vivo functioning of biomolecules. This work is a first step towards physically realistic in silico whole-cell models that connect molecular with cellular biology.

*For correspondence: sugita@riken.jp (YS); feig@msu.edu (MF)

**Competing interests:** The authors declare that no competing interests exist.

## Introduction

How biomolecules efficiently function in real biological environments with crowding and significant chemical and physical heterogeneity remains a fundamental question in biology (*Minton, 2001*). Typical cytoplasmic macromolecular concentrations are 300–450 g/L or 25–45 vol% (*Zimmerman and Trach, 1991*). Metabolites add about 10 g/L (*Bennett et al., 2009*). Volume exclusion upon crowding favors compact macromolecular states (*Minton, 2001*), but the full physico-chemical nature of cellular environments with attractive and repulsive interactions, solvation effects and co-solvents apparently leads to more varied effects (*Monteith et al., 2015*; *Harada et al., 2013*, *2012*; *Feig and Sugita, 2012*; *Kim and Mittal, 2013*; *Tanizaki et al., 2008*). The key question is whether and how the in vivo behavior of biological macromolecules differs from their well-characterized in vitro properties.

In-cell NMR and other recent experiments of cell-like solutions point at possible native state destabilization upon crowding (*Monteith et al., 2015*; *Inomata et al., 2009*; *Hong and Gierasch, 2010*; *Sakakibara et al., 2009*). Such observations contradict a simple excluded volume model (*Minton, 2001*) and a full understanding of how cellular environments modulate protein structures remains elusive. Cellular environments reduce the diffusive dynamics of macromolecules, but, again,

**eLife digest** Much of the work that has been done to understand how cells work has involved studying parts of a cell in isolation. This is particularly true of studies that have examined the arrangement of atoms in large molecules with elaborate structures like proteins or DNA. However, cells are densely packed with many different molecules and there is little proof that proteins keep the same structures inside cells that they have when they are studied alone.

To really understand how cells work, new ways to understand how molecules behave inside cells are needed. While this cannot be achieved directly, technology has now reached the stage where we can, to some extent, study living cells by recreating them virtually. Simulated cells can copy the atomic details of all the molecules in a cell and can estimate how different molecules might behave together.

Yu et al. have now developed a computer simulation of part of a cell from the bacterium, *Mycoplasma genitalium*, one of the simplest forms of life on Earth. This model suggested new possible interactions between molecules inside cells that cannot currently be studied in real cells. The model shows that some proteins have a much less rigid structure in cells than they do in isolation, whilst others are able to work together more closely to carry out certain tasks. Finally, the model predicted that small molecules such as food, water and drugs would move more slowly through cells as they become stuck or trapped by larger molecules.

These results could be particularly important in helping to improve drug design. Currently the simulations are limited, and can only model parts of simple cells for less than a thousandth of a second. However, in future it should be possible to recreate larger and more complex cells, including human cells, for longer periods of time. These could be used to better study human diseases and help to design new treatments. The ultimate goal is to simulate a whole cell in full detail by combining all the available experimental data.

details of how exactly macromolecules and metabolites move in an environment that is highly crowded and rich in varying interactions are unclear. Crowded environments also provide increased opportunities for weak protein-protein interactions due to frequent random encounters but it is unknown to what extent such weak interactions may benefit the efficiency of metabolic cascades or other coordinated biological processes.

As experiments are beginning to approach realistic cellular environments, it remains extremely challenging to probe biomolecular structure and dynamics in cellular environments without either perturbing the system that is being studied or the environment. Theoretical studies have the potential to overcome such challenges (*Im et al., 2016*). Whole-cell modeling based on the metabolic network of *Mycoplasma genitalium* (*MG*) has been able to predict phenotype variations (*Karr et al., 2012*), but without considering physical details. Molecular-level models have captured aspects of cellular environments (*McGuffee and Elcock, 2010*; *Ando and Skolnick, 2010*; *Cossins and Jacobson, 2011*), but the full biological complexity has not been reached (*Feig and Sugita, 2013*). Driven by data from high-throughput experiments, we built a comprehensive cytoplasmic model primarily on *MG* and its nearest relative, *Mycoplasma pneumoniae* (*Feig et al., 2015*; *Kühner et al., 2009*). Here, this model is subject to molecular dynamics simulations to examine in atomistic detail how realistic cellular environments affect the dynamic interplay of proteins, nucleic acids, and metabolites.

## Results

All-atom molecular dynamics (MD) simulations were applied to three atomistic cytoplasmic models containing proteins, nucleic acids, metabolites, ions and water, explicitly. We studied $MG_h$, based on a cytoplasmic model built previously with 103 million atoms in a cubic (100 nm) (*Bennett et al., 2009*) box (*Figure 1* and *Table 1*) (*Feig et al., 2015*), and two different subsections, $MG_{m1}$ and $MG_{m2}$, with 12 million atoms (*Table 1*). Unrestrained MD simulations were carried out for 20 ns ($MG_h$), 140 ns ($MG_{m1}$) (*Video 1*), and 60 ns ($MG_{m2}$). Although the simulation times, limited by resource constraints, may seem short, ensemble averaging over many copies of the same molecules

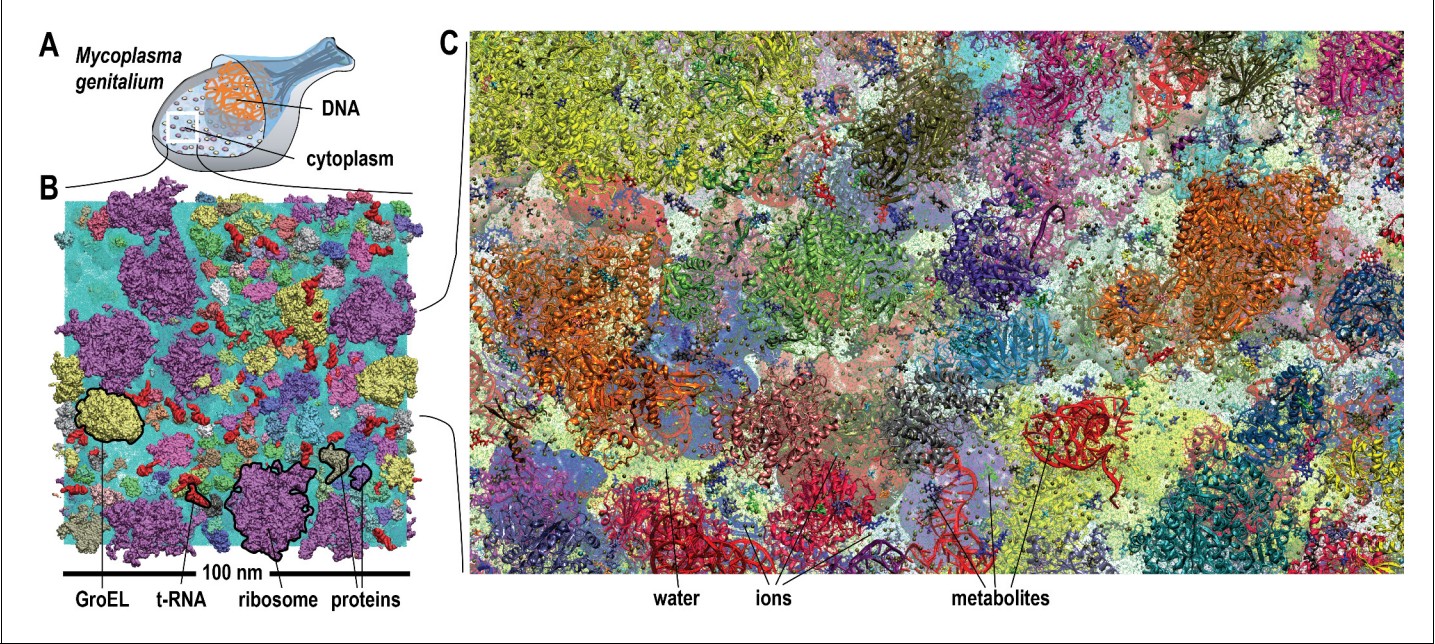

**Figure 1.** Molecular model of a bacterial cytoplasm. (**A**) Schematic illustration of *Mycoplasma genitalium* (*MG*). (**B**) Equilibrated $MG_h$ system highlighted with proteins, tRNA, GroEL, and ribosomes. (**C**) $MG_h$ cl ose-up showing atomistic level of detail. See also supplementary *Figures 1* and *2* for structures of individual macromolecules and metabolites as well as supplementary *Figure 3* for initial configurations of the simulated systems.

The following figure supplements are available for figure 1:

**Figure supplement 1.** Macromolecular components.

**Figure supplement 2.** Structure of metabolites in $MG_h$.

**Figure supplement 3.** Initial configurations of simulated systems.

in different local environments allowed for meaningful statistics. Furthermore, the three systems were started from different initial conditions providing further statistical significance.

## Native state stability of biomacromolecules in cellular environments

The stabilities of five proteins (phosphoglycerate kinase, PGK; pyruvate dehydrogenase E1.a, PDHA; NADH oxidase, NOX; enolase, ENO; and translation initiation factor 1, IF1) and tRNA (ATRN) in the cellular environments in terms of root mean square displacements (RMSD) from the initial homology models and radii of gyration ($R_g$) were compared with simulations in dilute solvents (*Figure 2A* and *Table 2*). We focused on these systems because of large copy numbers to obtain sufficient statistics. Average RMSD values with respect to the initial models were lower or the same in the cellular environment compared to dilute solvent for PGK, NOX, ENO, and IF1 but increased for PDHA. The stability of individual copies varied significantly presumably at least in part as a function of the local environment consistent with recent work by Ebbinghaus et al. that found significant variations in protein folding rates within a single cell (*Ebbinghaus et al., 2010*). Some copies of PDHA significantly departed from the native structures in the cellular environment and those molecules had extensive contacts with other proteins (*Figure 2—figure supplement 2* and *Video 2*) similar to previously observed destabilizations of native structures due to protein-protein interactions in crowded environments (*Harada et al., 2013*; *Feig and Sugita, 2012*). To further understand the mechanism by which PDHA became destabilized, we analyzed one copy that denatured significantly in more detail (we denote this copy as PDHA*). Time traces shown in *Figure 2—figure supplement 2* illustrate that the increase in RMSD coincides with the formation of protein-protein contacts, in particular with PYK. Additional energetic analysis indicates that the destabilization is driven by an overall decrease

**Table 1.** Simulated cytoplasmic systems.

| System | $MG_h$ | $MG_{m1}$ | $MG_{m2}$ | $MG_{cg}$ |
|---|---|---|---|---|
| Cubic box length (nm) | 99.8 | 48.2 | 48.2 | 106.2 |
| Program | GENESIS | GENESIS | NAMD | GENESIS |
| Simulation time | 20 ns | 140 ns | 60 ns | $10 \times 20$ µs |
| | | number of molecules | | |
| Ribosomes | 31 | 3 | 3 | 24 |
| GroELs | 20 | 3 | 3 | 24 |
| Proteins | 1238 | 182 | 133 | 1927 |
| RNAs | 284 | 28 | 44 | 298 |
| Metabolites | 41,006 | 5.005 | 5.072 | |
| Ions | 214,000 | 23,049 | 27,415 | |
| Waters | 26,263,505 | 2,944,143 | 2,893,830 | |
| Total # of atoms | 103,708,785 | 11,737,298 | 11,706,962 | |

See also **Figure 1—figure supplement 3** showing initial configurations and supplementary material with lists of the individual molecular components.

in the crowding free energy (see **Figure 2—figure supplement 2**). Further decomposition reveals a decrease in protein-protein electrostatic energies and van der Waals interactions while electrostatic solvation energies increase as PDHA becomes destabilized (**Figure 2—figure supplement 2E,F**). This means that favorable protein-protein electrostatic interactions between PDHA and the crowders are counteracted by unfavorable solvation as far as the dominant electrostatic component is concerned. The combination of the electrostatic and electrostatic solvation contributions increases (**Figure 2—figure supplement 2E**) suggesting that based on electrostatics and solvation alone the destabilization of PDHA* would not be favorable. However, this increase is more than outweighed by a decrease in the van der Waals interaction energy that suggests that, in the case of PDHA*, non-specific, shape-driven interactions ultimately lead to native state destabilization. In addition, the

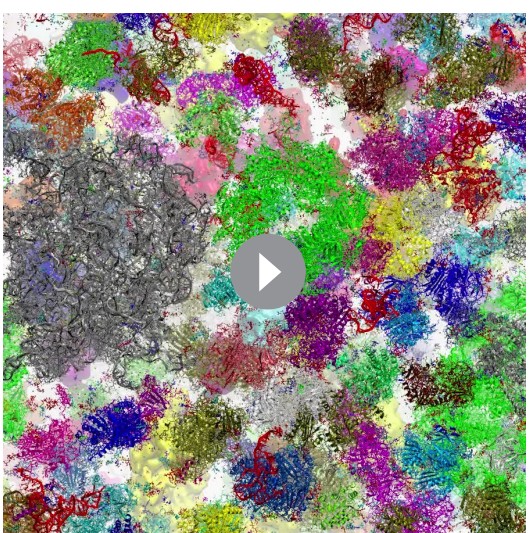

**Video 1.** Nanosecond dynamics of the $MG_{m1}$ system in atomistic detail. Macromolecules are shown with both cartoon and lines. Metabolites and ions are shown with stick or sphere. Macromolecules in back ground are shown with surface representation.

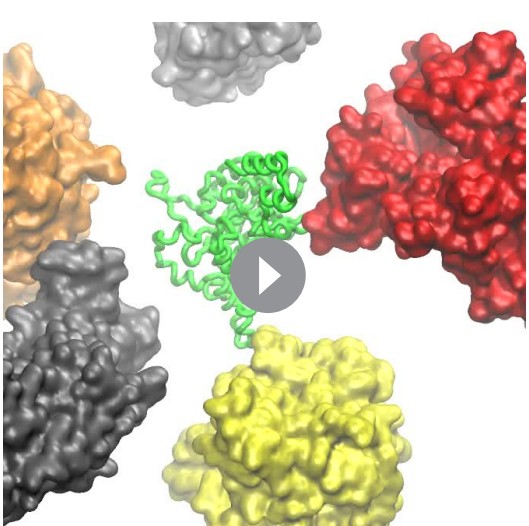

**Video 2.** Conformational dynamics highlighting partial denaturation of one copy of PDHA (green, tube) due to interactions with proteins in the vicinity.

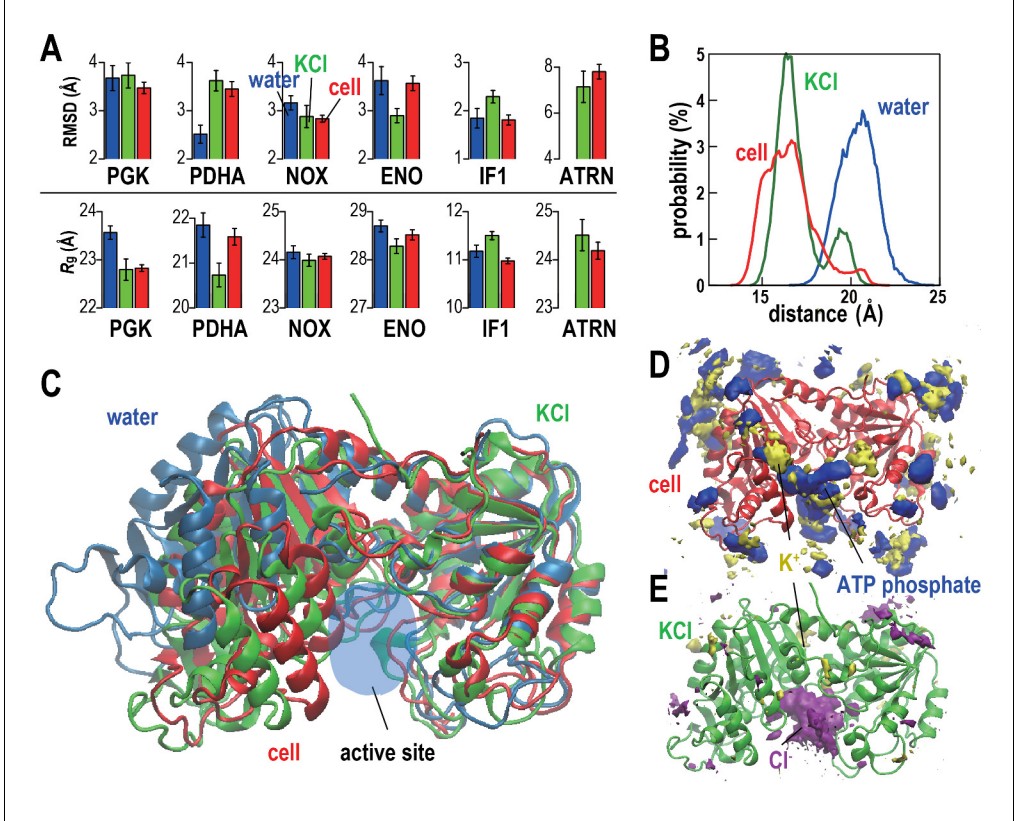

**Figure 2.** Conformational stability of macromolecules in crowded and dilute environments. (**A**) Time-averaged RMSDs (from starting structures) and radii of gyration ($R_g$) for selected macromolecules in $MG_{m1}$(red), in dilute solution with only counterions (blue) and with KCl excess salt (green). Statistical errors are with respect to copies of the same type. (**B**) Probability of the center of mass distances between the ligand binding sites $d_{lig}$ for PGK in $MG_{m1}$ (red), in water (blue), and in KCl (green). (**C**): Final snapshots of PGK in $MG_{m1}$ (red), in water (blue), and in KCl (green). (**D**) Time- and ensemble-averaged 3D distribution of atoms in the ATP phosphate group (blue, 0.002 $Å^{-3}$) and $K^+$ (yellow, 0.001 $Å^{-3}$) around PGK in $MG_{m1}$. (**E**) Time- and ensemble-averaged 3D distribution of $K^+$ (yellow, 0.001 $Å^{-3}$) and $Cl^-$ (purple, 0.001 $Å^{-3}$) around PGK in KCl aqueous solution. See also supplementary *Figures 1*, *2* and *3* showing time series of structural stability measures and the influence of the local crowding environment on the structure of PGK and PDHA.

The following figure supplements are available for figure 2:

**Figure supplement 1.** Time series of structural stability measures for selected macromolecules.

**Figure supplement 2.** Influence of local crowding environment on the structure of PDHA in $MG_{m1}$.

**Figure supplement 3.** Influence of metabolite binding and local crowding environment on the structure of PGK in $MG_{m1}$.

overall solvent-accessible surface area of the PDHA*-crowder system decreases as evidenced by the decrease in the asp term (which is proportional to the solvent-accessible surface area), further contributing to the interaction of the destabilized PDHA* with the crowder environment being more favorable than the initial non-interacting native PDHA*.

$R_g$ values, reflecting overall compactness, were generally lower in the cellular environment over dilute solvent as expected from the volume exclusion effect (*Minton, 2001*). However, dilute solvent with added KCl matching the molality of the cytoplasm led to a similar reduction in $R_g$ as in the cellular environment (*Figure 2A*). We focused additional analysis on PGK, where FRET measurements in the presence of polyethylene glycol (PEG) and coarse-grained simulations have suggested that its two domains come closer upon crowding concomitant with higher enzymatic activity (*Dhar et al., 2010*). In living cells, folded structures are also stabilized (*Ebbinghaus et al., 2010*; *Guo et al., 2012*). Consistent with these studies, the distance between the two ligand-binding sites ($d_{lig}$) in

**Table 2.** Simulated single protein reference systems.

| System | Cubic box [nm] | # of waters | # of ions | # of metabolites | # of atoms | Simulation time* [ns] |
|--------|---------------|-------------|-----------|------------------|------------|----------------------|
| PGK_w | 9.89 | 30,787 | Cl⁻: 8 | 0 | 98,886 | 4 × 140 |
| PGK_i | 9.87 | 30,374 | K⁺: 217, Cl⁻: 225 | 0 | 98,081 | 4 × 140 |
| PDHA_w | 9.90 | 31,032 | Na⁺: 7 | 0 | 98,779 | 2 × 140 |
| PDHA_i | 9.98 | 30,627 | K⁺: 224, Cl⁻: 217 | 0 | 97,998 | 2 × 140 |
| IF1_w | 9.92 | 32,785 | Cl⁻: 4 | 0 | 99,535 | 2 × 140 |
| IF1_i | 9.90 | 32,312 | K⁺: 233, Cl⁻: 237 | 0 | 98,582 | 2 × 140 |
| NOX_w | 9.89 | 30,473 | Cl⁻: 3 | 0 | 98,708 | 2 × 140 |
| NOX_i | 9.87 | 30,007 | K⁺: 222, Cl⁻: 225 | 0 | 97,754 | 2 × 140 |
| ENO_w | 9.85 | 28,050 | Na⁺: 2 | 0 | 98,330 | 2 × 140 |
| ENO_i | 9.84 | 27,648 | K⁺: 203, Cl⁻: 201 | 0 | 97,526 | 2 × 140 |
| ATRN_i | 9.88 | 31,734 | K⁺: 231, Cl⁻: 156 | 0 | 98,032 | 2 × 140 |
| ACKA_m | 14.71 | 102,379 | K⁺: 231, Cl⁻: 156 | 168 | 325,691 | 2 × 510 |

*The first 10 ns of each trajectory was discarded as equilibration.

$MG_{m1}$ decreased relative to that in water, but a similar decrease also occurred in the KCl solution (*Figure 2B–C*). The PGK domain cleft attracted high concentrations of ATP in the cell where Cl⁻ was found in the KCl simulations (*Figure 2D–E*). However, we found little correlation between $d$lig and the crowder coordination number of PGK ($N_c$) (*Figure 2—figure supplement 3*). To understand this observation in more detail, we looked again at one specific copy PGK (denoted as PGK*) where we correlated the time series of $d_{lig}$ with macromolecular crowder contacts, nucleotides entering the cleft, and the resulting additional charge (*Figure 2—figure supplement 3*). It can be seen that $d_{lig}$ closely tracks the charge in the cleft with more compact conformations occurring when more negative charge is present. The charge would screen electrostatic repulsion across the cleft between a large number of basic residues and allow the two ligand-binding domains to come closer. In this specific example, ATP and/or CTP entering the cleft region (*Figure 2—figure supplement 3E*) are responsible for bringing the negative charge to the cleft (*Figure 2—figure supplement 3E*). On the other hand, contacts with crowder molecules are not well correlated with $d_{lig}$ for this copy of PGK* (*Figure 2—figure supplement 3F*) mirroring the overall lack of correlation of $d_{lig}$ with crowder contacts (*Figure 2—figure supplement 3B*). These observations suggest that electrostatic stabilization by ions can induce similar effects as may be expected due to volume exclusion effects and that non-specific interactions with metabolites can affect biomolecular structures in unexpected ways.

## Weak non-specific interactions of metabolically related enzymes

The high concentration of macromolecules in the cytoplasm allows macromolecules to weakly interact without forming traditional complexes. Such 'quinary' interactions have been proposed before (*Monteith et al., 2015*), but experimental studies in complex cellular environments are challenging and their biological significance is unclear. Based on distance changes for proximal macromolecular pairs relative to the initially randomly setup systems ($\Delta d_{AB}$) we compared interactions between regular proteins, RNAs, and huge complexes (ribosome and GroEL) to determine relative affinities for each other between these types of macromolecules (*Figure 3A*). The electrostatically-driven strong repulsion among RNAs and between RNAs and huge macromolecules (mainly ribosomes) is readily apparent. Repulsion between proteins and RNA and huge complexes is weaker, whereas protein-protein interactions were neutral. However, proteins involved in the glycolysis pathway showed weak attraction. An attraction of glycolytic enzymes is consistent with experimental data indicating the formation of dynamic complexes to enhance the multi-step reaction efficiency via substrate channeling (*Dutow et al., 2010*). Specific complex formation or specific weak interactions may allow related enzymes to associate, but here we observe *non-specific* weak associations that do not follow identifiable interaction patterns between different enzymes and require an alternate rationalization.

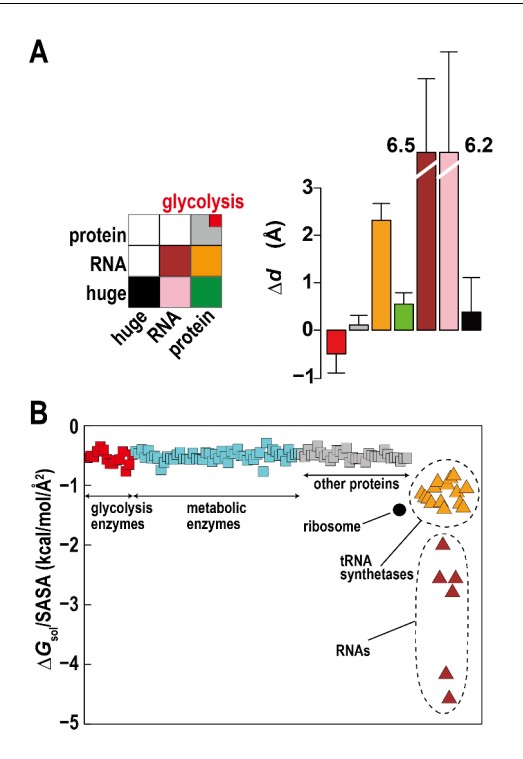

**Figure 3.** Association of metabolic proteins in crowded environments. (**A**) Intermolecular distance changes between initial and final time ($\Delta d_{AB}$) for pairs of glycolytic enzymes, other regular proteins, RNAs, and ribosomes/ GroEL (huge). (**B**) Solvation free energies $\Delta G_{sol}$ normalized by the solvent-accessible surface area (SASA) for equilibrated copies of macromolecules in $MG_{m1}$ using GBMV (*Lee et al., 2003*) in CHARMM (*Brooks et al., 2009*). See also supplementary *Figure 1* showing the influence of large macromolecules on the association of small proteins based on simple Lennard-Jones mixtures.

The following figure supplement is available for figure 3:

**Figure supplement 1.** Influence of large macromolecules on the association of small proteins.

---

One explanation can be found based on the relative solvation free energies of glycolytic enzymes. Calculated solvation free energies for glycolytic enzymes are similarly favorable as other proteins, but they are less favorable compared to tRNA, aminoacyl tRNA synthetases with tRNA, and ribosomes, which together make up about a third of the macromolecular mass (*Figure 3B*). This suggests that solvation effects effectively cause a weak attraction of glycolytic enzymes (and other similar proteins) as they are relatively hydrophobic compared to the RNA containing molecular components.

Another aspect is the large size difference between glycolytic enzymes and ribosomes. Simulations of two-component mixtures of Lennard-Jones spheres to focus on entropic effects (*Figure 3— figure supplement 1*) show an increased concentration of smaller particles within a distance of two to three times their radii when large particles are present vs. a homogenous mixture of small particles. This is an indirect consequence of Asakura-Oosawa-type depletion forces where attraction between large particles excludes smaller particles bringing them closer to each other (*Asakura and Oosawa, 1958*). Therefore, the presence of large ribosomes enhances the proximity of smaller enzymes.

We note that a possible biological significance of relative size differences between enzymes and other smaller biomolecules such as metabolites has been raised before by Srere (*Srere, 1984*) in the context of protein-protein interactions and substrate channeling in metabolically-related enzymes. However, these early ideas were not yet informed by the full knowledge of structural biology that is

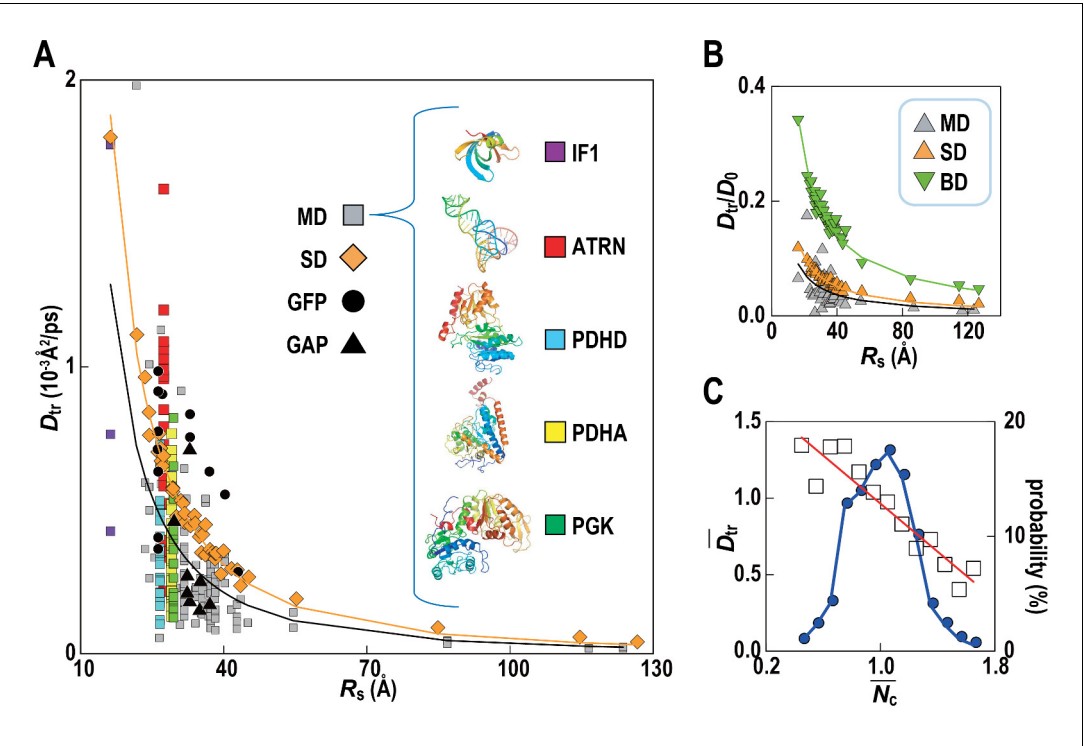

**Figure 4.** Translational diffusion of macromolecules in $MG_{m1}$ slows down as a function of Stokes radius and is dependent on local crowding. (**A**) Translational diffusion coefficients ($D_{tr}$) of macromolecules in $MG_{m1}$ vs. Stokes radii ($R_s$) from MD, and SD compared with experimental data for green fluorescent protein (GFP) and GFP-attached proteins in *E. coli* (**Nenninger et al., 2010**). Fitted functions are $D_{tr} = 341/R_s^2$ (MD) and $D_{tr} = 496/R_s^2$ (SD). (**B**) $D_{tr}/D_0$ using $D_0$ from HYDROPRO (**Fernandes and de la Torre, 2002**) for $MG_{m1}$ (grey), SD (orange), and BD (green). Fitted functions for $D_{tr}/D_0$ are $1.5/R_s$ (MD), $2.0/R_s$ (SD), and $5.6/R_s$ (BD). (**C**) Normalized translational diffusion coefficient ($\overline{D_{tr}}$) vs. normalized coordination number ($\overline{N_c}$) for selected macromolecules (white squares) and distribution of macromolecules vs. $\overline{N_c}$ (blue line). See also supplementary **Figures 1** and **2** showing the dependency of the calculated diffusion coefficients on the observation time and the influence of the local crowding environment on the diffusion coefficients of individual proteins.

The following figure supplements are available for figure 4:

**Figure supplement 1.** Dependency of translational diffusion coefficient $D_{tr}$ on the maximum observation time $\tau_{max}$.

**Figure supplement 2.** Influence of local crowding environment on $D_{tr}$.

available today and, therefore, did not provide clear physical rationales for how enzymes sizes and size distributions may relate to biological function.

## Diffusive properties of biological macromolecules in cellular environments

Translational diffusion coefficients ($D_{tr}$) of Green Fluorescent Proteins (GFPs) and GFP-attached proteins (GAPs) are reduced about tenfold in *Escherichia coli* cells compared to dilute solutions (**Nenninger et al., 2010**) but much less is known how exactly the slow-down in diffusion depends on the local cellular environment. We calculated $D_{tr}$ for the macromolecules in $MG_{m1}$ as a function of their Stokes radius, $R_S$, from our simulations (**Figure 4**; **Video 3**). Remarkably, experimental values are matched without adjusting parameters suggesting that our model based on *MG* may capture the physical properties of bacterial cytoplasms more generally. Convergence analysis from our data (**Figure 4—figure supplement 1**) combined with previous studies of diffusion rates (**McGuffee and**

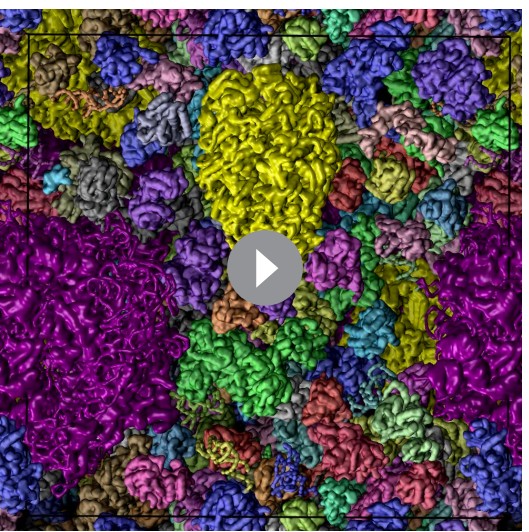

**Video 3.** Diffusive motion of macromolecules during the last 130 ns of the $MG_{m1}$ system. Macromolecules are shown with surface representation. Ribosomes and GroELs are colored violet and yellow respectively. Other groups of molecules are colored differently for each individual macromolecule.

*Elcock, 2010*; *Ando and Skolnick, 2010*) suggests that long-time diffusion rates are approached already at 100 ns, although with slight overestimation (*McGuffee and Elcock, 2010*). There is also excellent agreement between the all-atom MD simulations and estimates of $D_{tr}$ from coarse-grained Stokesian dynamics (SD) simulations of spherical macromolecules ($MG_{cg}$, *Table 1*) in the presence of hydrodynamic interactions (*Ando and Skolnick, 2010*) (*Figure 4B*). The ratio $D_{tr}/D_0$ that describes the slow-down in diffusion due to crowding relative to diffusion in dilute solvent $D_0$, based on values estimated by HYDROPRO (*Fernandes and de la Torre, 2002*), decreases as $1/R_s$ as expected from previous studies of diffusion in crowded solutions (*McGuffee and Elcock, 2010*; *Roosen-Runge et al., 2011*; *Szymański et al., 2006*; *Banks and Fradin, 2005*). Given the classical $1/R_s$ dependency in dilute solvent, $D_{tr}$ follows a $1/R_s$ (*Zimmerman and Trach, 1991*) dependency in crowded environments. Inverse quadratic functions are excellent fits to both the atomistic and coarse-grained simulation results (*Figure 4A*).

Although the ensemble-averaged diffusive properties follow a simple $1/R_s$ (*Zimmerman and Trach, 1991*) function, there is a wide spread of $D_{tr}$ in different copies of the same macromolecule type (*Figure 4A*) as a consequence of experiencing different local environments (*Figure 4—figure supplement 2*). The diffusion constant $D_{tr}$ as a function of the normalized coordination number with surrounding macromolecules, $\overline{N_c}$, follows a linear trend when averaged over different types of macromolecules (*Figure 4C*) with diffusion rates, on average, varying three-fold between environments with the least and most contacts with surrounding molecules. As molecules diffuse through the cytoplasm, a given molecule thus exhibits a spatially varying rate of diffusion over time scales of 1 μs–1 ms, based on how long it takes for the smallest and largest macromolecules to diffuse by twice their Stokes radius.

We also report rotational motion (*Figure 5*) from our simulations. Rotational properties of macromolecules in physically realistic cellular environments have not yet been described in detail due to simplified models and the use of spherical approximations in past studies. We find that, in general, rotational diffusion follows the same trend as for translational diffusion, including a very similar dependency on local crowding (*Figure 5—figure supplement 1*). A similar reduction of translational and rotational diffusion upon crowding on shorter, sub-microsecond time scales found here is consistent with experimental data from quasi-elastic neutron backscattering and NMR relaxometry (*Roosen-Runge et al., 2011*; *Roos et al., 2016*). However, our simulations are too short to probe the suggested protein species dependent decoupling of rotational and translational diffusion on longer time scales based on pulsed field gradient NMR measurements of dense protein solutions (*Roos et al., 2016*).

## Diffusive properties of solvent and metabolites in cellular environments

As expected (*Harada et al., 2012*), the diffusion of water and ions was slowed down significantly in the cytoplasmic environment (*Table 3*), but little is known about the behavior of low molecular weight organic molecules in the cytoplasm. Translational diffusion rates $D_{tr}$ of the metabolites in $MGm1$ exhibit a much more rapid decrease with increasing molecular weight, proportional to $1/R_s$ (*Bennett et al., 2009*), compared with the $1/R_s$ (*Zimmerman and Trach, 1991*) decrease seen for macromolecules (*Figure 6A*). Especially highly-charged phosphates diffused much slower than would be expected simply due to crowding. This observation is in stark contrast to recent experimental

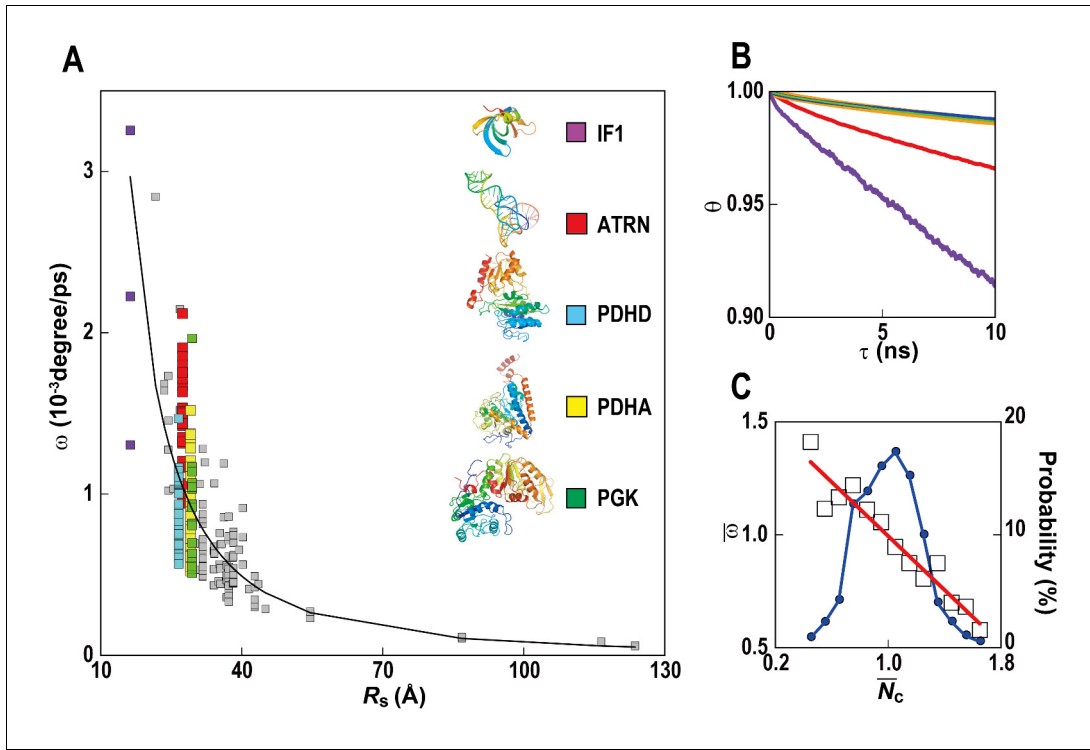

**Figure 5.** Rotational diffusion of macromolecules. (**A**) Averaged angular velocity ($\omega$) of macromolecules in $MG_{m1}$ as a function of their Stokes radii ($R_s$) (gray squares with IF1, ATRN, PDHD, PDHA, and PGK highlighted in purple, red, blue, yellow, and green, respectively) (**B**) Rotational correlation functions ($\theta$) of macromolecules (IF1, ATRN, PDHD, PDHA, and PGK colored in purple, red, blue, yellow, and green, respectively). (**C**) Normalized angular velocities ($\overline{\omega}$) vs. normalized coordination numbers ($\overline{N_c}$) (white square) averaged over abundant macromolecules vs. macromolecular distribution as in **Figure 4**. See also supplementary **Figure 1** showing the influence of the local crowding environment on the rotational diffusion of individual macromolecules.

The following figure supplement is available for figure 5:

**Figure supplement 1.** Influence of local crowding environment on angular velocity ω.

results (*Rothe et al., 2016*) that suggest that the diffusion of small molecules should be reduced less in crowded environments than for the much larger macromolecules. Based on our simulations this is a consequence of a large fraction of the metabolites interacting non-specifically with macromolecules (*Figure 6B*). Once metabolites are bound on the surface of a macromolecule, diffusion becomes two-dimensional and slows down considerably as illustrated in detail for ATP and valine

**Table 3.** Diffusion of water and ions. Translational diffusion constants [Å²/ps] in the cytoplasm (Mgm1) and dilute solvent (simulation of PGK in excess salt matching cytoplasmic concentration).

|  | Cytoplasm | | Dilute solvent | |
| --- | --- | --- | --- | --- |
|  | $\tau_{max}$ 1.0 (ns) | $\tau_{max}$ 10 (ns) | $\tau_{max}$ 1.0 (ns) | $\tau_{max}$ 10 (ns) |
| water | 0.32 | 0.29 | 0.42 | 0.41 |
| $K^+$ | 0.079 | 0.068 | 0.22 | 0.21 |
| $Na^+$ | 0.017 | 0.015 | N/A | N/A |
| $Cl^-$ | 0.17 | 0.14 | 0.22 | 0.21 |
| $Mg^{2+}$ | 0.0073 | 0.0051 | N/A | N/A |

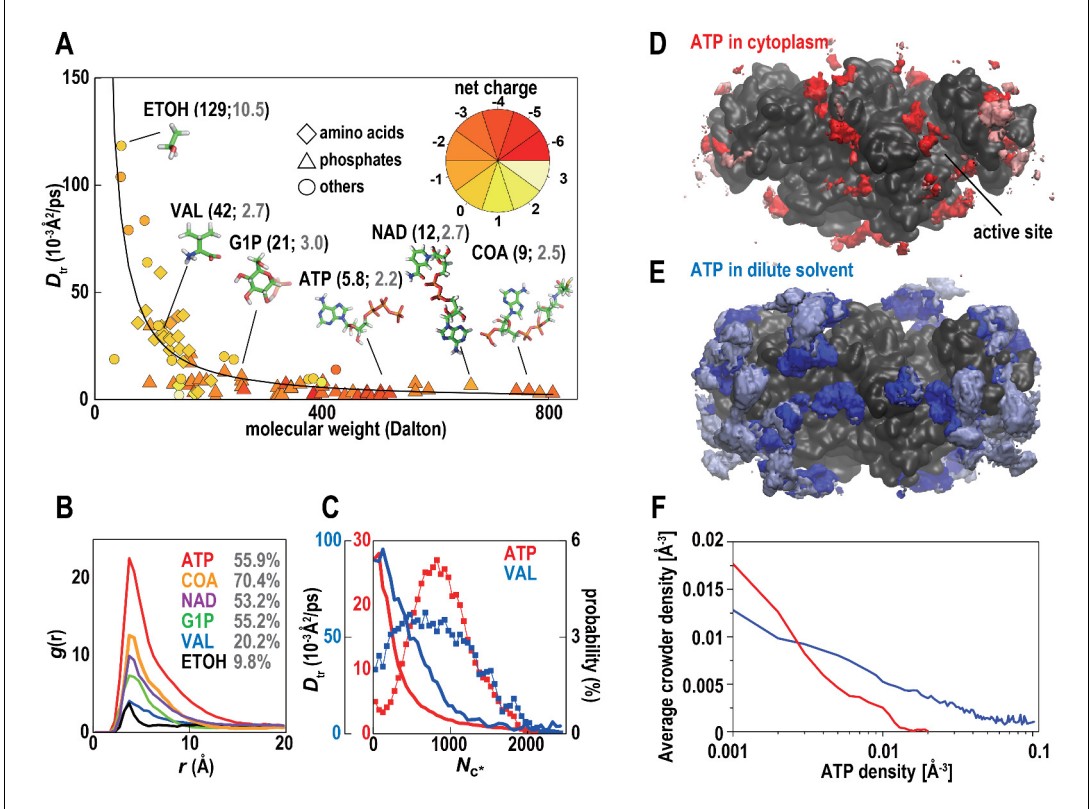

**Figure 6.** Metabolites in cytoplasmic environments interact extensively with macromolecules resulting in significantly reduced diffusion. (A) Translational diffusion coefficients ($D_{tr}$) for metabolites in $MGm1$ as a function of molecular weight (phosphates: diamond; amino acids: triangles; others: circles; color reflects charge). For abundant metabolites, diffusion coefficients in bulk (black) and during macromolecular interaction (grey) are given in parentheses. (B) Normalized conditional distribution function, $g(r)$, for heavy atoms of selected metabolites vs. the distance to the closest macromolecule heavy atom. The percentage of metabolites interacting with a macromolecule is listed. (C) $D_{tr}$ of ATP and VAL as a function of the coordination number with macromolecules ($N_{c*}$) (line) and the distribution of $N_{c*}$ (%) (line with points). (D) Time-averaged 3D distribution of all atoms in ATP (red, 0.008 Å$^{-3}$) around ACKA molecules in $MG_{m1}$. Pink color indicates regions where all-atom crowder densities also exceed 0.008 Å$^{-3}$. (E) Same as in (D) but the density of ATP is shown in dilute solvent (blue) with light blue indicating overlap with the crowder density distribution form the $MG_{m1}$ simulations. (F) Correlation between average crowder atom densities in $MG_{m1}$ and volume density grid voxel ATP densities in dilute (blue) and crowded (red) environments. In the dilute case, we compute the crowder atom densities in $MGm1$ as a function of the grid ATP densities in the dilute simulations of PDHA. Therefore, high average crowder atom densities in the cytoplasmic model at sites with high ATP densities under dilute conditions means that those ATP sites would be displaced by interacting crowders in the cytoplasmic environment. See also supplementary *Figure 1* showing analysis details for the calculation of the ATP distributions.

The following figure supplement is available for figure 6:

**Figure supplement 1.** ATP distribution in cytoplasmic environments.

(*Figure 6C*; *Video 4*). Trapping of metabolites on macromolecular surfaces reduces the effective concentration of freely-diffusing metabolites consistent with recent experiments that have inferred a large fraction of surface-interacting metabolites due to crowding (*Duff et al., 2012*). A recent analysis of absolute metabolite concentrations in *E. coli* has found that most concentrations were above the values of the Michaelis constant $K_m$ for enzymes binding those metabolites (*Bennett et al., 2009*). This has led to the conclusion that most enzyme active sites should be saturated under biological conditions. However, this argument neglects the possibility of significant non-specific metabolite-protein interactions suggested by the present study, which would imply much lower active site occupancies than expected from the absolute metabolite concentrations.

We further compared the interaction of ATP with the highly abundant ATP-binding protein acetate kinase (ACKA) between cellular and dilute environments with ATP present at the same molality.

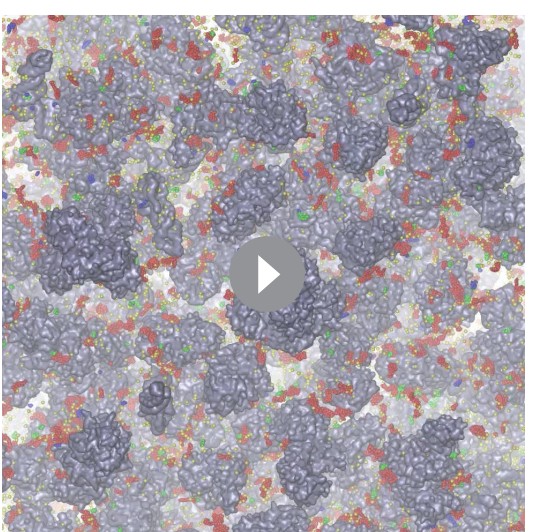

**Video 4.** Diffusive motion of metabolites during the last 130 ns of the $MG_{m1}$ system. Macromolecules are shown with surface representation. Metabolites and ions are shown with van der Waals spheres. Phosphates, amino acids, ions, and other metabolites are highlighted with red, green, yellow, and blue.

The number density profiles $\rho(r)$ show a decreased concentration of ATP in the vicinity of ACKA (*Figure 6—figure supplement 1D*) in the cellular environment. In part, this results from reduced accessible volume around ACKA upon crowding (*Figure 6—figure supplement 1E*), but competition can play another role since ATPs can interact with many other proteins instead of ACKA. The three-dimensional distribution of ATP around ACKA (*Figure 6D*) shows not just reduced binding of ATP in the cell but also binding at different sites. For example, ATP binding to the cleft region between the two dimer subunits only features prominently in the cellular environment. This is a consequence of crowder interactions competing with ATP binding (*Figure 6*). A large fraction of the high-density ATP binding sites seen in dilute solvent overlap with crowder interaction sites while only very few ATP binding sites under crowded conditions overlap with crowder atom densities (*Figure 6D/E/F*). The active site cleft is largely unaffected by crowder interactions and therefore binding to the active site should be unaffected by crowding. However, because the non-active site binding sites are very different we expect that the binding kinetics is affected by the presence of the crowders. Further exploration of how exactly the thermodynamics and kinetics of metabolite-protein interactions are affected by the presence of the cytoplasm will require extensive additional analysis and will be deferred to a future study.

## Discussion

This study provides unprecedented details of the interactions of biomolecules in a complete cytoplasmic environment. Our model emphasizes that in vivo environments are significantly different from in vitro conditions and illustrates that the inclusion of the full physical environment and the presence of all cellular components exerts more than just the simple volume exclusion effect commonly associated with crowding. We find new evidence for native state perturbations in cellular environments as a result of 'quinary' protein-protein interactions consistent with recent NMR studies (*Monteith et al., 2015*) but also as a result of electrostatic factors due to interactions with ions and metabolites that compete with volume exclusion effects. Our results suggest that native-state perturbations towards functionally compromised states under in vivo conditions may be a more general phenomenon that should be taken into account when interpreting in vitro studies and relying on structures obtained via crystallography.

Another significant insight that is of biological importance is the observation of weak association of glycolytic enzymes in our model. While glycolytic enzymes dominate our cytoplasmic model, both of our explanations, reduced solvation energies and entropic sorting due to size differences, would apply to the majority of metabolic enzymes and we expect that metabolic enzymes as a whole are brought into closer proximity in the cellular environment, thereby generally enhancing metabolic rates. An intriguing question is whether biological systems have evolved to exhibit such characteristics. The large relative size of ribosomes and their large RNA contents may therefore be desirable features outside the immediate context of their function in translation.

Our simulations also allowed us to carry out a detailed analysis of diffusive properties that revealed significant heterogeneity in macromolecular diffusion as a function of the local environment and a surprisingly significant reduction of metabolite diffusion as a result of sticky interactions with

macromolecular surfaces. The latter has implications for the number of metabolites actually present in bulk solvent and has consequences for the mechanism of ligand binding in cellular environments.

One recurring aspect in our study is the prominent role of electrostatics that is manifested in different forms, such as the differential solvation between metabolic proteins and RNAs and RNA-containing complexes and metabolite-protein interactions that alter protein structure via electrostatic screening but are also at least in part responsible for reducing the amount of freely-diffusing metabolites. These findings mirror in part ideas by Spitzer and Poolman that hypothesized electrochemical effects to be a major factor in organizing crowded cytoplasms (*Spitzer and Poolman, 2005*).

The effect of cellular environments on and by metabolites and metabolic enzymes is a major focus of the present study. The present work points at reduced effective ligand concentrations near the macromolecule surface and altered protein-ligand interactions under cellular conditions. Therefore, cell-focused in silico drug design protocols that capture competing interactions and altered kinetic properties under cellular conditions could offer significant advantages over current single-molecule protocols.

All of the results presented here were obtained via computer simulations that, although increasingly reliable and based on experimentally-driven models, still only represent theoretical predictions. One potential concern is that the structural models used here were obtained largely via homology modeling. While the accuracy of structure prediction has increased over the last decade (*Kryshtafovych et al., 2014*), using such models may affect the accuracy of our results reported here, in particular with respect to the effects of the cellular environment on protein stability. In order to diminish such effects, we focused our analysis on relative comparisons of the same homology models in the cellular environment and in dilute solvent which we believe is meaningful even if the models are only approximations of the real structures. At the same time, the analysis of diffusive properties and non-specific protein-protein interactions is expected to be driven mostly by overall shape, size, and electrostatics rather than specific structural details and, therefore, we expect those results not to be affected significantly by the use of homology models.

The simulations presented here are limited by relatively short simulation lengths due to computational resource constraints. The observed macromolecular diffusion was largely limited to motion within a local environment without enough time to trade places with other macromolecules and certainly without exploring a substantial fraction of the volumes of our simulation systems. The short simulation times affect the estimates of long-time diffusive behavior including the possibility of anomalous diffusion that would not be expected to appear until macromolecules actually exchange with each other. On the other hand, the analysis of native state destabilization and non-specific protein-protein interactions relies essentially on interactions within just the local environment that are being sampled extensively on the time scale of our simulations, while averaging over many different copies in different environments is assumed to provide equivalent statistics to following a single copy that diffuses over much longer time scales. We therefore expect that much longer simulations would primarily reduce the statistical uncertainties without fundamentally altering the results reported here with respect to macromolecular stability and interactions. However, we expect that longer time scales would allow sampling of specific macromolecular and ligand binding equilibria, e.g. tRNA binding to tRNA synthetases, protein oligomerization, or binding of metabolite substrates to their respective target protein active sites, none of which we observed in the course of our simulations.

Force field inaccuracies are another concern and this work is a true test of the transferability and compatibility of the CHARMM force field parameters for close molecular interactions under conditions where force field parameters have not been validated extensively and artifacts, e.g. overstabilization of protein-protein contacts, are possible (*Petrov and Zagrovic, 2014*). Therefore, additional simulations with other force fields are desirable and follow-up by experiments is essential. The suggested metabolite-induced compaction of PGK should be observable experimentally and it should also be possible to measure weak associations of metabolic enzymes in dense protein solutions with and without nucleic acids and with and without ribosomes using suitable fluorescence probes, for example. Variations of diffusive properties as a function of the local cellular environments and the diffusion of metabolites in cellular environments may be more difficult to determine experimentally but we hope that our work will stimulate experimental efforts to determine such properties.

The next step from the model presented here is a whole-cell model in full physical detail to include the genomic DNA, the cell membrane with embedded proteins, and cytoskeletal elements.

Such a model would require in excess of 10 (*Tanizaki et al., 2008*) particles at an atomistic level of detail. As additional experimental data becomes available and computer platforms continue to increase in scale, this may become possible in the foreseeable future. Such a whole-cell model would bring to bear the tremendous advances in structural biology to make a complete connection between genotypes and phenotypes at the molecular level that is difficult to achieve with the empirical systems biology models in use today.

## Materials and methods

### Model system construction

We constructed a comprehensive cytoplasmic model at pH = 7 based on *MG* containing proteins, nucleic acids, metabolites, ions, and explicit water, consistent with predicted biochemical pathways as described previously (*Feig et al., 2015*). The model is meant to cover a cytoplasmic section that does not contain membranes, DNA, or cytoskeletal elements and includes only essential gene products. Molecular concentrations were estimated based on proteomics and metabolomics data for a very closely related organism, *M. pneumoniae* and macromolecular structures were predicted via homology modeling and complexes were built where possible (see *Figure 1*).

### All-atom molecular dynamics simulations

The cytoplasmic model covered a cubic box of size (100 nm) (*Bennett et al., 2009*) with about 100 million atoms ($MG_h$; *Figure 1A*). This system corresponds to 1/10th of a whole *MG* cell. Based on $MG_h$, we also built two smaller subsets ($MG_{m1}$ and $MG_{m2}$), each with a (50 nm) (*Bennett et al., 2009*) volume and containing about 12 million atoms. The subsets were constructed like the complete system but using molecular copy numbers from two different 1/eighth subsets of the $MG_h$ system. All-atom MD simulations were carried out over 140 ns for $MG_{m1}$ and 60 ns for $MG_{m2}$ of with the first 10 ns were discarded as equilibration. For $MG_h$, we performed 20 ns MD simulation with the first 5 ns considered as equilibration. $MG_{m1}$ and $MG_h$ trajectories were obtained with GENESIS (*Jung et al., 2015*) on the K computer. $MG_{m2}$ was run as a control using NAMD 2.9 (*Phillips et al., 2005*). Analysis was performed with the in-house program MOMONGA and the MMTSB Tool Set (*Feig et al., 2004*). System details are given in *Table 1* and a list of macromolecules and metabolites are provided as supplementary material.

Systems with single macromolecules in explicit solvent were built for phosphoglycerate kinase (PGK), pyruvate dehydrogenase E1.a (PDHA), NADH oxidase (NOX), enolase (ENO), translation initiation factor 1 (IF1), tRNA (ATRN), and acetate kinase (ACKA). PGK, PDHA, NOX, ENO and IF1, were solvated in pure water (with counterions) and aqueous solvent with excess KCl. The molality of $K^+$ ions was adjusted to match the $MG_{m1}$ system. ATRN was only simulated in the presence of the added salt. ACKA was simulated in water and in a mixture of metabolites with the components and molalities chosen such that they match the $MG_{m1}$ system. Details for these systems are given in *Table 2*. MD simulations of single macromolecules in dilute solvent were repeated two to four times using different initial random seeds. Details about the number and length of runs for each system are given in the *Table 2*.

In all atomistic simulations, initial models were minimized for 100,000 steps via steepest descent. For the first 30 ps of equilibration, a canonical (*NVT*) MD simulation was performed with backbone $C_\alpha$ and P atoms of the macromolecules harmonically restrained (force constant: 1.0 kcal/mol/Å [*Zimmerman and Trach, 1991*]) while gradually increasing the temperature to 298.15 K. We then performed an isothermal-isobaric (*NPT*) MD simulation for 10 ns without restraints. Production MD runs were carried out under the *NVT* ensemble for $MG_{m1}$ and $MG_{m2}$. For $MG_h$, we ran a total of 20 ns in the *NPT* ensemble without switching to the *NVT* ensemble. The CHARMM c36 force field (*Best et al., 2012*) was used for all of the proteins and RNAs. The force-field parameters for the metabolites were either taken from CGenFF (*Vanommeslaeghe et al., 2010*) or constructed by analogy to existing compounds. All bonds involving hydrogen atoms in the macromolecules were constrained using SHAKE (*Ryckaert et al., 1977*). Water molecules were rigid by using SETTLE (*Miyamoto and Kollman, 1992*) which allowed a time step of 2 fs. Van der Waals and short-range electrostatic interactions were truncated at 12.0 Å, and long-range electrostatic interactions were calculated using particle-mesh Ewald summation (*Darden et al., 1993*) with a (512) (*Bennett et al.,*

2009) grid for $MG_{m1}$ and $MG_{m2}$ and a (1024) (Bennett et al., 2009) grid for $MG_h$. The temperature (298.15 K) was held constant via Langevin dynamics (damping coefficient: 1.0 ps$^{-1}$) and pressure (1 atm) was regulated in the NPT runs by using the Langevin piston Nosé-Hoover method (Hoover et al., 2004; Nosé, 1984) (damping coefficient: 0.1 ps$^{-1}$).

## Brownian and Stokesian dynamics simulations

A coarse-grained model of the MG cytoplasm, $MG_{cg}$, was built for Stokesian and Brownian dynamics simulations. Here, each macromolecule was represented by a sphere with the radius $a$ set to the Stokes radius estimated by HYDROPRO (Fernandes and de la Torre, 2002) based on the modeled structures. The number of copies for each macromolecule was set to be 8 times larger than that in $MG_{m1}$ because most of the atomistic simulation data presen ted here is based on this system. The number and radii of macromolecules are listed in supplementary material.

$MG_{cg}$ was simulated via Brownian dynamics (BD) without hydrodynamics interactions (HIs) (Ermak and McCammon, 1978) and Stokesian dynamics (SD) (Brady and Bossis, 1988; Durlofsky et al., 1987), which includes not only the far-field HI but also the many-body and near-field HIs. For BD simulations without HIs, we used a second-order integration scheme introduced by Iniesta and de la Torre (Iniesta and García de la Torre, 1990), which is based on the original first-order algorithm developed by Ermak and McCammon (Ermak and McCammon, 1978). We only considered repulsive interactions between particles to take into account excluded volume effects, which are described by a half-harmonic potential,

$$V_{ij} = \begin{cases} \frac{1}{2}k(r_{ij} - a_i - a_j - \Delta)^2 & \text{if } r_{ij} < a_i + a_j + \Delta \\ 0 & \text{if } r_{ij} \geq a_i + a_j + \Delta \end{cases} \tag{1}$$

where $k$ is the force constant, $r_{ij}$ is the distance between particles $i$ and $j$, $a_i$ and $a_j$ are the radii of particles $i$ and $j$, respectively, and $\Delta$ is an arbitrary parameter representing a buffer distance between particles. In this study, a $\Delta = 1\text{Å}$ and $k = 10k_BT/\Delta^2$ with the Boltzmann constant $k_B$ were used, which means that $V_{ij} = 5k_BT$ at the distance $r_{ij} = a_i + a_j$. For SD simulations, the modified mid-point BD algorithm introduced by Banchio and Brady (Banchio and Brady, 2003) and based on Fixman's idea (Fixman, 1978) was used. All BD and SD simulations were performed under periodic boundary conditions at 298 K. A time step of 8 ps was used, which roughly corresponds to $0.0005 \times a^2/D_{tr}$ for the particle with the smallest radius in the system, where $D_{tr}$ is the translational diffusion constant (which is equal to $k_BT/6\pi\eta a$ with the viscosity of water $\eta$). Ten independent simulations were performed, each over 20 µs with different random seeds from randomly generated different initial configurations, using BD and SD implementations in the program GENESIS (Jung et al., 2015).

## Calculation of root mean square displacements (RMSD) of macromolecules

RMSD values were determined for $C_\alpha$ and P atoms after best-fit superpositions. Structures obtained after short-time (10 ps) MD simulations in water started from the initial predicted models were used as reference structures since experimental structures are not available.

Highly flexible regions where $C_\alpha$ and P atoms had root mean square fluctuations (RMSF) larger than 3.0 Å$^2$ (for proteins) or 4.0 Å$^2$ (for tRNA) were eliminated from the analysis. Time and copy-averaged values with their respective standard errors were calculated from $t_0$ = 50 ns to $t_{end}$ = 130 ns in $MG_{m1}$.

## Calculation of translational diffusion coefficients

The time evolution of the square displacement of a macromolecule $\alpha$ in a given time window $i$ ($r^2(\alpha, i, \tau)$) was obtained by tracking the center of mass of $\alpha$. Multiple profiles of $r^2(\alpha, i, \tau)$ were obtained by sliding windows up to a size of $\tau_{max}$ = 10 ns using an interval of $\Delta t_i$ = 10 ps for macromolecules and up to $\tau_{max}$ = 1 ns for metabolites using an interval of $\Delta t_i$ = 1 ns starting from the beginning of the production trajectories up to $t_{end} - \tau_{max}$, where $t_{end}$ is the maximum length of a given simulation (see Table 1). In the case where diffusion coefficients are compared with coordination numbers (see below) $\Delta t_i$ = 500 ps was chosen. These profiles were then averaged to obtain mean square displacements (MSD) according to:

$$\langle r^2(\alpha,\tau)\rangle_t = \frac{1}{(t_{\text{end}} - \tau_{\text{max}})/\Delta t_i} \sum_i r^2(\alpha, i, \tau) \tag{2}$$

To obtain translational diffusion coefficient $D_{\text{tr}}$, a linear function was fitted to the MSD curve and $D_{\text{tr}}$ was computed from the slope of the fitting line using the Einstein relation.

$$D_{\text{tr}} = \frac{\langle r^2(\alpha,\tau)\rangle_t}{6\tau} \tag{3}$$

For *Figures 4A*, *5A* and *6A* and *Figure 4—figure supplement 1*, only the last 80% of the MSD curve were used for fitting to enhance the accuracy. The entire MSD curve was used to generate *Figures 4C*, *5C* and *6C*, and *Figure 4—figure supplement 2* and *Figure 5—figure supplement 1*.

The simulation results were shown with experimentally measured diffusion coefficients of green fluorescence protein (GFP) and GFP-attached proteins (*Nenninger et al., 2010*). In order to map the experimental data onto the simulations results, Stokes radii, $R_s$, of GFPs (GFP, GFP oligomers, and GFP attached proteins) were estimated from the relation between the molecular weight, $M_w$, and $R_s$ obtained by HYDROPRO (*Fernandes and de la Torre, 2002*) for macromolecules in $MG_{m1}$. $M_w$ vs $R_s$ data were fitted with an exponential function ($R_s = 2.54\ M_w^{2.86}$) which was used to estimate $R_s$ for the GFP constructs based on their $M_w$ values.

### Analysis of rotational motion

To analyze the overall tumbling motion of a macromolecule $\alpha$, we adopted the procedure developed by Case et al. (*Wong and Case, 2008*) using the rotation matrix that minimizes the RMSD of $\alpha$ against the reference structure, the rotational correlation function in a given time window $i$ ($\theta(\alpha, i, \tau)$) as a function of $\tau$ was obtained using sliding windows as in the calculation of the translational diffusion coefficients (see above) as follows with $\tau_{\text{max}} = 10$ ns:

$$\langle \theta(\alpha,\tau)\rangle_t = \frac{1}{(t_{\text{end}} - \tau_{\text{max}})/\Delta t_i} \sum_i \theta(\alpha, i, \tau)\, (\tau < \tau_{\text{max}}) \tag{4}$$

Time-ensemble averages of rotational correlation functions for macromolecule type $A$ were obtained by taking average for multiple copies of $\alpha$ belonging to the type $A$.

$$\langle \theta(A,\tau)\rangle_{\alpha t} = \frac{1}{N} \sum_\alpha \langle \theta(\alpha,\tau)\rangle_t \quad (\alpha \in A) \tag{5}$$

The rotational relaxation time $\tau_{\text{rel}}$ was obtained by fitting a single exponential (*McGuffee and Elcock, 2010*)

$$\langle \theta(A,\tau)\rangle_{\alpha t} \propto \exp(-\tau/\tau_{\text{rel}}) \tag{6}$$

Finally, the rotational diffusion coefficient of macromolecule type $A$ was obtained as

$$D_{\text{rot}}(A) = 1/2\tau_{\text{rel}} \tag{7}$$

To obtain time-averaged angular velocities for a molecule $\alpha$, the inner product of the rotated unit vectors at $t = t_i$ and $t = t_i + \tau_{\text{max}}$ were calculated as:

$$\langle \Delta \mathbf{e}_j(\tau_{\text{max}})\rangle_t = \frac{1}{(t_{\text{end}} - \tau_{\text{max}})/\Delta t_i} \sum_{t_i} \langle \mathbf{e}_j(t_i + \tau_{\text{max}}) \cdot \mathbf{e}_j(t_i)\rangle_j \tag{8}$$

The time-averaged angular velocity $\langle \omega \rangle_t$ of $\alpha$ in units of degrees was obtained as follows,

$$\langle \omega \rangle_t = \frac{180}{\pi} \arccos\left(\frac{\langle \Delta \mathbf{e}_j(\tau_{\text{max}})\rangle_t}{\tau_{\text{max}}}\right) \tag{9}$$

### Calculation of coordination number of crowders

To measure the local degree of crowding around a given target molecule $\alpha$, we used the number of backbone $C_\alpha$ and P atoms in other macromolecules within the cutoff distance $R_{\text{cut}} = 50$ Å from the

closest $C_\alpha$ and P atoms of $\alpha$ at a given time $t$ as the *instantaneous coordination number* of crowder atoms, $N_c(\alpha, t)$, (For metabolites, we calculated the instantaneous coordination number of heavy atoms in crowder from the *center of mass* of a target metabolite $m$ with a cutoff value of $R_{cut}$ = 25 Å. This quantity is denoted as $N_{c*}(m, t)$). Time averages of $N_c(\alpha, t)$, and $N_{c*}(m, t)$ were calculated over 10 ns windows advanced in 500 ps steps for macromolecules and over 1 ns windows advanced 1 ns steps for metabolites, respectively.

## Characterization of macromolecular interactions

Macromolecular interactions were analyzed by using the center of mass distance for macromolecule pairs. The change of the distance between a target macromolecule $\alpha$ and one of the surrounding macromolecule $\beta$, $\Delta d_{\alpha\beta}$, during the entire production trajectory from $t_0$ to $t_{end}$ was calculated as:

$$\Delta d_{\alpha\beta}(R_{cut}) = \langle r_c(\alpha, \beta, t_{end}) \rangle_t - \langle r_c(\alpha, \beta, t_0) \rangle_t, \tag{10}$$

where $\langle \rangle_t$ denotes the time average of center of mass distance $r_c(\alpha, \beta, t)$ in the short time window $\tau_{short}$ at the beginning and at the end of the time window. The selection of surrounding molecules $\beta$ was based on the scaled distances $\bar{r}$ between two protein pairs

$$\bar{r}(\alpha, \beta) = \frac{2r_c(\alpha, \beta)}{R_s(\alpha) + R_s(\beta)} \tag{11}$$

where $R_s(\alpha/\beta)$ is the Stokes radius of each molecule. $\beta$ was selected as surrounding molecule when the time-averaged distance from $\alpha$ is shorter than the cutoff distance $R_{cut}$ at the beginning of time window.

$$\langle \bar{r}(\alpha, \beta, t_i) \rangle_t < R_{cut}. \tag{12}$$

The ensemble average of the distance change between two macromolecule groups $A$ and $B$ as a function of the cutoff radius, $R_{cut}$, $\Delta d_{AB}(R_{cut})$, was obtained for macromolecule pairs belonging to each group. In this study, $\Delta d_{AB}(R_{cut})$ was calculated using the longest time window for $MG_{m1}$ ($t_{end}$ = 130 ns, $\tau_{short}$ = 5 ns), $MG_{m2}$ ($t_{end}$ = 50 ns, $\tau_{short}$ = 5 ns), and $MG_h$ ($t_{end}$ = 15 ns, $\tau_{short}$ = 0.5 ns). The profile at $R_{cut} \approx 2$ reflects the short-range interaction (picking up the macromolecule pairs which are almost fully attached each other), while it converges to zero at larger $R_{cut}$ because the number of macromolecule pairs having no interaction rapidly increase. $\Delta d_{AB}(R_{cut})$ was then averaged between $R_{cut}$ = 2 and 3 to reduce the noise. The averaged value is denoted simply as $\Delta d_{AB}$. A cutoff distance of $R_{cut}$ = 3 corresponds to macromolecule pairs separated by about their diameter. $\Delta d_{AB}$ values were calculated for $MG_h$, $MG_{m1}$ and $MG_{m2}$, and combined with a weighted average according to the different lengths of the trajectories. $\Delta d_{AB}$ were obtained for the half of macromolecule pairs (whose scaled distance are initially less than 3.0) selected randomly from each macromolecule group. We repeated this calculation for 50 times, and obtained standard deviations (SD) for $50 \times 2$ = 100 values. Standard error was obtained by $SD/\sqrt{2}$.

Macromolecular association was analyzed separately for protein-protein, protein-RNA, and RNA-RNA interactions (see text). We separately analyzed interactions among proteins involved in the glycolysis pathways, which consist of HPRK (HPr/HPr kinase/phosphorylase), PYK (pyruvate kinase), TPIA (triosephosphate isomerase), GAPA (glyceraldehyde-3-phosphate dehydrogenase), PFKA (6-phosphofructokinase), FBA (fructose-biphosphate aldolase), ENO (enolase), PGI (glucose-6-phosphate isomerase), PGM (phosphoglycerate mutase) and PGK (phosphoglycerate kinase). Both multimeric and monomeric units were included in the analysis.

## Calculation of spatial distribution functions of metabolites

Proximal radial distribution functions, $g(r)$, were calculated for the distances between centers of heavy atoms in target metabolites and the nearest heavy atom of surrounding macromolecules. The number of atoms in a given target metabolite, which exist in the theoretically accessible volume ($V(r)$, show as gray layers in *Figure 6—figure supplement 1A*), $n(r)$, were obtained as a function of distance $r$. The volume and pairwise distances were averaged from snapshots taken at 5 ns intervals for the cellular systems and at 1 ns intervals for the dilute systems. The atomic number density $\rho(r)$ was

calculated by dividing $n(r)$ by $V(r)$. To obtain the normalized distribution functions, $\rho(r)$ were divided by the atomic number density in the furthest region from the surface of macromolecules $\rho(\infty)$,

$$g(r) = \frac{n(r)}{V(r)\rho(\infty)}. \tag{13}$$

To obtain the numerical values for $V(r)$, the periodic boundary box was divided into 1 Å grids, and $V(r)$ were approximated by counting the grids whose center is inside the $V(r)$ (thick black squares in *Figure 6—figure supplement 1A*). The profiles of $g(r)$ were obtained in a histogram with bin size 0.5 Å. $\rho(\infty)$ was approximated by taking the average value of $\rho(r)$ from 20 Å to 25 Å.

$\rho(r)$ of ATP was also obtained only with respect to the 13 ACKA molecules in the crowded system $MG_{m1}$ as well as for the single ACKA under dilute conditions with metabolites and ions ($ACKA_{\_m}$) (*Figure 6—figure supplement 1D*). We used the entire 130 ns production trajectory of $MG_{m1}$. For dilute conditions ($ACKA_{\_m}$) a total of 1 µs sampling from multiple trajectories was used.

In addition to $\rho(r)$ and $g(r)$, the three-dimensional distribution of the atomic number density $\langle\rho(\mathbf{r})\rangle$ of metabolites or ions around the target proteins were generated with a grid size of 1 Å. $\langle\rho(\mathbf{r})\rangle$ were calculated for each snapshot, and iso-density surfaces were projected onto a reference structure of the target protein by removing translational and rotational motion of the protein (*Figure 2C–E*, and 6D-E in the main manuscript). For the density calculations, snapshots saved at 100 ps intervals for cellular systems and saved at 50 ps for dilute systems were used.

Because ACKA has two symmetrical domains (i.e., homodimer), ATP atoms around one domain were transposed to the other domain, and $\langle\rho(\mathbf{r})\rangle$ were then calculated by counting both original and transposed solvent atoms in the same grid to generate symmetry-averaged densities (see *Figure 6D–E*).

## Characterization of the two-dimensional diffusion of metabolites

For selected metabolites we analyzed the interaction with macromolecule surfaces. Metabolites were considered to be interacting with a macromolecular surface if the distance between the center of mass of a metabolite and the nearest heavy atom of any of the surrounding macromolecules was less than 10 Å for the large metabolites COA and NAD, and less than 8 Å for ATP, VAL, G1P, and ETOH. If a metabolite interacted continuously with the same macromolecule for more than 5 ns before and after a given time, we considered the metabolite to be moving on the macromolecular surface, therefore exhibiting two-dimensional diffusion. Mean square displacements (MSD) of these metabolites were averaged separately and the slope was divided by four instead of six when determining $D_{tr}$ to reflect two-dimensional vs. three-dimensional diffusion.

## Acknowledgements

We thank K Takahashi and T Yanagida for discussion and Mr. Takase for technical assistance with computations. Computational resources were used at the RIKEN Advanced Institute for Computational Science (K computer) through the HPCI strategic research project (Project ID: hp120309, hp130003, hp140229, hp150233, hp160207), at the Texas Advanced Computing Center through an XSEDE allocation (TG-MCB090003), and at the RIKEN Integrated Cluster of Clusters (RICC). Funding was provided by the Fund from the High Performance Computing Infrastructure (HPCI) Strategic Program of the Ministry of Education, Culture, Sports, Science and Technology (MEXT) (to YS), a grant from Innovative Drug Discovery Infrastructure through Functional Control of Biomolecular Systems, Priority Issue 1 in Post-K Supercomputer Development (to YS), a Grant-in-Aid for Scientific Research on Innovative Areas "Novel measurement techniques for visualizing 'live' protein molecules at work" (No. 26119006) (to YS), a grant from JST CREST on "Structural Life Science and Advanced Core Technologies for Innovative Life Science Research" (to YS), RIKEN QBiC and iTHES (to YS), a Grant-in-Aid for Scientific Research (C) from MEXT (No. 25410025) (to IY), and from NIH (GM092949, GM084953) (to MF), and NSF (MCB 1330560) (to MF).

# Additional information

## Funding

| Funder | Grant reference number | Author |
|---|---|---|
| National Institutes of Health | R01 GM092949 | Michael Feig |
| National Science Foundation | MCB 1330560 | Michael Feig |
| National Science Foundation | XSEDE TG-MCB090003 | Takaharu Mori Michael Feig |
| Ministry of Education, Culture, Sports, Science, and Technology | High Performance Computing Infrastructure Strategic Program | Yuji Sugita |
| Ministry of Education, Culture, Sports, Science, and Technology | Innovative Drug Discovery Infrastructure through Functional Control of Biomolecular Systems | Yuji Sugita |
| Ministry of Education, Culture, Sports, Science, and Technology | Grant-in Aid 26119006 | Yuji Sugita |
| Ministry of Education, Culture, Sports, Science, and Technology | Grant-in Aid 25410025 | Isseki Yu |
| Japan Science and Technology Agency | CREST | Yuji Sugita |
| RIKEN | | Isseki Yu Takaharu Mori Tadashi Ando Ryuhei Harada Jaewoon Jung Yuji Sugita Michael Feig |
| National Institutes of Health | R01 GM084953 | Michael Feig |

The funders had no role in study design, data collection and interpretation, or the decision to submit the work for publication.

## Author contributions

IY, TA, MF, Conception and design, Acquisition of data, Analysis and interpretation of data, Drafting or revising the article; TM, Acquisition of data, Analysis and interpretation of data, Drafting or revising the article; RH, JJ, Acquisition of data, Drafting or revising the article; YS, Conception and design, Analysis and interpretation of data, Drafting or revising the article

## Author ORCIDs

Michael Feig, http://orcid.org/0000-0001-9380-6422

# Additional files

## Supplementary files

• Supplementary file 1. Detailed lists of system components. List of Macromolecules. Copy numbers for each macromolecule (represented by tag name) in four simulation systems. The Stokes radius $R_s$ is given in the last column. Groups and types of metabolites. Net charge and number of copies for each metabolite (represented by tag name) in three simulation systems. Phosphates are highlighted with a pink background.

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
