## [Decision Letter]

Thank you for submitting your article "Macromolecular Dynamics, Stability, and Atomistic Interactions in a Bacterial Cytoplasm" for consideration by *eLife*. Your article has been favorably evaluated by John Kuriyan (Senior Editor) and three reviewers, one of whom, Yibing Shan (Reviewer #1), is a member of our Board of Reviewing Editors.

The reviewers have discussed the reviews with one another and the Reviewing Editor has drafted this decision to help you prepare a revised submission. It should be possible for you to address these issues without running new simulations. Note, also, that the editor asks you to change the title of the manuscript so that it conveys to the reader more clearly what was actually done.

Summary:

This pioneering work simulates a substantial fraction of a bacterial cytoplasm using all-atom molecular dynamics simulations, aiming to elucidate how interactions in a crowded environment affect the stability and diffusion properties of macromolecules. It is a technical feat to construct and to simulate such an enormous all-atom model of more than 100 million atoms. The simulations break new ground of molecular dynamics in terms of system scale. The work shows that, among other findings, protein structural dynamics and ligand binding in the crowded environment differ from diluted solution as theoretically expected and that metabolic enzymes exhibit a tendency to cluster spatially.

Essential revisions:

The reviewers recognize the herculean effort and the groundbreaking nature of this work. To publish this work in *eLife*, however, the manuscript needs revisions in several aspects:

1) The simulation lengths of tens of nanoseconds are too short for the system to equilibrate and to have a chance to depart significantly from the initial model. Different macromolecules will, on average, experience different environments which will affect their behavior (indeed, the authors observe wildly different behaviors for different copies of the same molecules). Given the very small diffusion coefficients involved (~1*10-3 A2/ps for macromolecules and ~1*10-2 A2/ps for ions), the macromolecules move on average by a few angstroms to a few nanometers during the entire simulation. Given the size of the systems 50-100 nm, they are largely "frozen" in such a short timescale. Ideally, the simulation lengths should be of micro- to milliseconds, allowing enough time for the macromolecules to diffuse the entire length of the simulation box. The reviewers understand that such simulation length is impossible even with today's largest supercomputers, but the manuscript needs to acknowledge this important limitation of the work and discuss in-depth its implications to help the readers correctly interpret the results.

2) The manuscript also needs to discuss the potential issues of inaccuracy of the models the simulations are initiated from. For example, the fact that many of the protein structures the system entails are homology models with only short equilibration (20 picoseconds) will likely have consequences for the findings concerning protein stability. It is difficult to assess the "stability" of these protein models from the RMSD traces, in particular on such short timescale. It is likely that some of these models would undergo substantial unfolding and rearrangements on longer timescales as suggested by the relatively large RMSDs from the stating structure, even when flexible regions are omitted. These potential issued should acknowledged and analyzed to the extent that is feasible.

3) More details should be given concerning how non-specific intermolecular interactions affect protein conformational dynamics. It will be useful to follow one or two protein molecules as examples, with a focus on the intermolecular interactions they experienced in the course of the simulation.

4) The mechanism by which inter-biomolecular interactions affect protein ligand binding is obscure from the manuscript. Why is ATP distribution on the ACKA molecule differs in cytoplasm and in aqueous solution? The reason for this somewhat surprising observation needs further Discussion.

5) The manuscripts and the potential impact of this work would benefit if the authors can make an effort to improve the text. The conclusions need to be clearly stated in the Abstract and the Discussion. For instance, in the Abstract, the authors write about nonspecific interactions 'affecting' several properties. The reader needs to know what is affected, by how much and compared to what. This lack of clarity is a general feature. Similarly, the authors write about 'stability' in the first paragraph of the subsection “Native state stability of biomolecules in cellular environments”, but never define it.

The authors write about decreased distances between proteins in their model, but the reader needs to know, 'compared to what?' For a baseline, the authors could refer to Spitzer and Poolman paper: 2005 Electrochemical structure of the crowded cytoplasm. Trends Biochem Sci 30:536.

---

## [Author Response]

*Essential revisions:*

The reviewers recognize the herculean effort and the groundbreaking nature of this work. To publish this work in eLife, however, the manuscript needs revisions in several aspects:

1) The simulation lengths of tens of nanoseconds are too short for the system to equilibrate and to have a chance to depart significantly from the initial model. Different macromolecules will, on average, experience different environments which will affect their behavior (indeed, the authors observe wildly different behaviors for different copies of the same molecules). Given the very small diffusion coefficients involved (~1*10-3 A2/ps for macromolecules and ~1*10-2 A2/ps for ions), the macromolecules move on average by a few angstroms to a few nanometers during the entire simulation. Given the size of the systems 50-100nm, they are largely "frozen" in such a short timescale. Ideally, the simulation lengths should be of micro- to milliseconds, allowing enough time for the macromolecules to diffuse the entire length of the simulation box. The reviewers understand that such simulation length is impossible even with today's largest supercomputers, but the manuscript needs to acknowledge this important limitation of the work and discuss in-depth its implications to help the readers correctly interpret the results.

The point is well-taken. One way we are addressing this issue is by combining analysis for three different systems (to have different initial configurations). We are also taking advantage of ensemble averaging and focused our analysis of macromolecules and metabolites on those that are present at large copy numbers. So, effectively, we would argue, we already have sampling for some molecules on the microsecond time scale (by combining sampling from different copies). This is already mentioned in the text.

Nevertheless, we followed the reviewers’ suggestion and added further discussion in the Discussion section.

2) The manuscript also needs to discuss the potential issues of inaccuracy of the models the simulations are initiated from. For example, the fact that many of the protein structures the system entails are homology models with only short equilibration (20 picoseconds) will likely have consequences for the findings concerning protein stability. It is difficult to assess the "stability" of these protein models from the RMSD traces, in particular on such short timescale. It is likely that some of these models would undergo substantial unfolding and rearrangements on longer timescales as suggested by the relatively large RMSDs from the stating structure, even when flexible regions are omitted. These potential issued should acknowledged and analyzed to the extent that is feasible.

The initial equilibration of each model was short but there was additional equilibration after constructing the cytoplasmic models. The results concerning protein stability were reported only for the last 80 ns of the longest trajectory (neglecting the first 50 ns as equilibration). Furthermore, we fully recognize the issues with using homology models but to control for that, we focused our analysis on the relative comparison between the cellular and dilute solvent environments where we simulated the same homology models. We believe that the relative comparison is meaningful even for approximate structural models of the proteins in our system.

We added further discussion in the sixth paragraph of the Discussion section to elaborate on this point.

3) More details should be given concerning how non-specific intermolecular interactions affect protein conformational dynamics. It will be useful to follow one or two protein molecules as examples, with a focus on the intermolecular interactions they experienced in the course of the simulation.

We carried out additional analyses to provide more insight into how the cellular environment affects proteins. We selected PDHA and PGK as two examples. For PDHA, we show time traces for one example where increases in RMSD and Rg are correlated with specific crowder contacts and changes in intra- and inter-molecular energy components to better illustrate the driving forces at play. For PGK, we show that crowder contacts, although present, are less important than the presence of charge in the active site cleft to favor more compact states. Again this is shown via time traces for a selected copy of PGK. A description of the results from the additional analyses was added to the main text with Figure 2—figure supplement 1,Figure 2—figure supplement 2 now showing the additional data.

*4) The mechanism by which inter-biomolecular interactions affect protein ligand binding is obscure from the manuscript. Why is ATP distribution on the ACKA molecule differs in cytoplasm and in aqueous solution? The reason for this somewhat surprising observation needs further Discussion.*

The short answer is that the presence of crowder molecules occludes binding sites present in dilute solvent. To show this we now compare averaged densities of crowder atoms with ATP around ACKA. We believe that there is a direct effect of simply displacing ATP and an indirect effect of ATPs binding to different sites compared to dilute solvent because not all of the dilute solvent sites are available. Interestingly, the active site is buried deeply enough to be unaffected by direct overlap with crowder molecules but based on the differences in binding sites we expect that ligand diffusion towards the active site is strongly affected by the crowded environment. These are questions that would warrant extensive further analysis, but to keep the present manuscript focused, we added only brief additional discussion to the main text with additional data shown in Figure 6.

5) The manuscripts and the potential impact of this work would benefit if the authors can make an effort to improve the text. The conclusions need to be clearly stated in the Abstract and the Discussion. For instance, in the Abstract, the authors write about nonspecific interactions 'affecting' several properties. The reader needs to know what is affected, by how much and compared to what.

We rewrote the Abstract to state our findings more clearly.

This lack of clarity is a general feature. Similarly, the authors write about 'stability' in the first paragraph of the subsection “Native state stability of biomolecules in cellular environments”, but never define it.

We added text to clarify what we mean by stability.

The authors write about decreased distances between proteins in their model, but the reader needs to know, 'compared to what?' For a baseline, the authors could refer to Spitzer and Poolman paper: 2005 Electrochemical structure of the crowded cytoplasm. Trends Biochem Sci 30:536.

The reference is with respect to the initial models that were setup by random placements. What we are focusing on is the relative change of different types of macromolecules moving relatively further away from each other while others come closer. We added text to clarify this point. We also added some additional discussion to the Discussion section to emphasize electrostatic effects and refer to the related ideas discussed in the Spitzer & Poolman paper.